# Pd-based bimetallic catalysts for HET-PHIP

Dudari B. Burueva[1,2], Aleksandr Y. Stakheev[3], Igor V. Koptyug[1,2]

[1]Laboratory of Magnetic Resonance Microimaging, International Tomography Center, SB RAS, Novosibirsk, 630090, Russia
[2]Novosibirsk State University, Novosibirsk, 630090, Russia
[3]N.D. Zelinsky Institute of Organic Chemistry, RAS, Moscow, 119991, Russia

*Correspondence to*: Igor V. Koptyug (koptyug@tomo.nsc.ru)

**Abstract.** Production of hyperpolarized catalyst-free gases and liquids by heterogeneous hydrogenation with parahydrogen (HET-PHIP) can be useful for various technical as well as biomedical applications, including in vivo studies, investigations of mechanisms of industrially important catalytic processes, enrichment of nuclear spin isomers of polyatomic gases, and more. In this regard, the wide systematic search for heterogeneous catalysts effective in pairwise $H_2$ addition required for the observation of PHIP effects is crucial. Here in this work we demonstrate the competitive advantage of Pd-based bimetallic catalysts for HET-PHIP. The dilution of catalytically active Pd with less active Ag or In atoms provides the formation of atomically dispersed $Pd_1$ sites on the surface of Pd-based bimetallic catalysts, which are significantly more selective toward pairwise $H_2$ addition compared to the monometallic Pd. Furthermore, the choice of the dilution metal (Ag or In) has a pronounced effect on the efficiency of bimetallic catalysts in HET-PHIP, as revealed by comparing Pd-Ag and Pd-In bimetallic catalysts.

## 1. Introduction

Among the nuclear spin hyperpolarization (HP) techniques that are currently becoming very popular in NMR and MRI, parahydrogen-based approaches such as PHIP and SABRE are of particular interest (Adams et al., 2009; Duckett and Mewis, 2012; Green et al., 2012; Iali et al., 2018; Reineri et al., 2015). They are relatively easy to implement while at the same time they compete successfully with other powerful HP techniques such as DNP (Ardenkjaer-Larsen, 2016; Jannin et al., 2019; Lumata et al., 2015; Rossini, 2018; Zhao et al., 2018). Since the first report of the PHIP effect (Bowers and Weitekamp, 1987), the majority of such studies are based on the homogeneous catalysis by transition metal complexes in solution. Many such catalysts can achieve pairwise addition of $H_2$ to a substrate and/or to a catalytically active center, which is usually a prerequisite for parahydrogen-based hyperpolarization. At the same time, there are several significant reasons why PHIP via heterogeneous hydrogenations (HET-PHIP) (Balu et al., 2009; Gutmann et al., 2010; Kaltschnee et al., 2019; Kovtunov et al., 2013, 2016, 2020a) can be advantageous. In particular, in vivo applications certainly require catalyst-free hyperpolarized fluids. Removal of potentially toxic transition metal complexes utilized in the homogeneous implementations of PHIP and SABRE, while possible (Cavallari et al., 2020; Kidd et al., 2018; Knecht et al., 2020), still remains one of the primary

challenges to be solved. In this respect, solid heterogeneous catalysts can be filtered out much faster, and thus higher polarization levels can be available upon injection. HET-PHIP is also directly applicable to the production of HP gases (Ariyasingha et al., 2020; Kovtunov et al., 2020b; Salnikov et al., 2019), which appears promising as a means to provide inhalable HP contrast agents for in vivo use in lung MRI. Furthermore, heterogeneous processes are most suited for the production of continuous streams of catalyst-free HP fluids (Kovtunov et al., 2014), which is expected to be useful for various technical as well as biomedical applications.

A separate yet strongly related field of research is the application of HP substances in the studies of chemical processes. In particular, PHIP has proven useful in the mechanistic studies of homogeneous catalytic reactions that involve $H_2$ (Duckett and Sleigh, 1999). In this respect, extension of the parahydrogen-based NMR signal enhancement to the mechanistic studies of heterogeneous hydrogenations and related processes (Du et al., 2020; Salnikov et al., 2015, 2018; Zhivonitko et al., 2016) is highly desirable. This is because heterogeneous catalysts and processes by far predominate in the modern large-scale industrial catalysis, while at the same time they are significantly more sophisticated by their nature. Developing enhanced analytical tools for such studies is a challenging but rewarding task.

Finally, the HET-PHIP approach can prove useful for addressing yet another challenge of modern science, namely the production/enrichment of nuclear spin isomers of polyatomic molecules (NSIM). As mentioned above, the addition of p-$H_2$ to a substrate or intermediate usually has to be pairwise for the correlated spin state of the nascent H atoms to be carried over to a reaction product or intermediate. However, if the two H atoms remain magnetically equivalent there, their correlated spin state, even if preserved, is not immediately revealed as an NMR signal enhancement. This has been demonstrated in numerous studies in solution in the context of the long-lived spin states (Levitt, 2019). Furthermore, hydrogenation of symmetric gaseous substrates is a potential route to the enrichment of their NSIM, as successfully demonstrated earlier for ethylene in a heterogeneous hydrogenation of acetylene with parahydrogen (Zhivonitko et al., 2013). This development is rather significant – while there are several known methods for enrichment or separation of NSIM of polyatomic molecules (Chapovsky and Hermans, 1999; Kilaj et al., 2018; Krüger et al., 2018), to date none of them is able to produce sufficient quantities of an enriched polyatomic gas for NMR signal enhancement applications, including the reported production of the singlet spin state of $^{15}N_2$ by SABRE in solution (Bae et al., 2018). Thus, the work of Zhivonitko et al. (2013) so far remains the only demonstration of NMR signal enhancement derived from a NSIM-enriched polyatomic gas.

Based on the major recent progress achieved in p-$H_2$-based hyperpolarized NMR, it can be reasonably expected that potential utilization of NSIM-enriched polyatomic molecules can advance this field of research and practice much further, by significantly expanding the scope of reactions and substrates/products for the production of HP agents for in vivo use as well as extending the advanced mechanistic studies to new classes of highly important catalytic processes. This, however, does not exhaust the list of activities that would benefit from the facile production/enrichment of NSIM of polyatomic molecules. In modern science, of significant interest are the properties of NSIM, including their interconversion processes (Chapovsky and Hermans, 1999) and NSIM behavior upon phase transitions (Curl et al., 1966; Hama et al., 2018), the NSIM-related selection rules in molecular spectroscopy (Kanamori et al., 2017; Ozier et al., 1970) and upon chemical transformations (Hu

et al., 2020; Kilaj et al., 2018; Oka, 2004), NSIM in astrophysics and astrochemistry research (Hama and Watanabe, 2013; Hily-Blant et al., 2018; Shinnaka et al., 2016; Tielens, 2013), and more.

To date, however, most heterogeneous catalysts demonstrated a limited efficiency in the pairwise hydrogen addition, or in some cases the low yields of the desired product, or both (Balu et al., 2009; Gutmann et al., 2010; Kaltschnee et al., 2019; Kovtunov et al., 2013, 2016, 2020a). Better heterogeneous catalysts are thus required, something which is constantly searched for in heterogeneous catalysis in general as more active and selective industrial catalysts have a major impact on all areas of our life. One of the powerful recent trends is the exploration of the so-called single-site and single-atom heterogeneous catalysts (Liu et al., 2018; Samantaray et al., 2020), which often outperform other catalyst types in terms of product selectivity and can ensure a much more efficient use of active metals, reducing the costs and contributing to cleaner chemical industry. Selectivity issues are highly important in essentially all catalytic processes; in particular, this includes semihydrogenation of alkynes in the presence of alkenes without converting them into alkanes.

One promising type of catalysts for selective hydrogenation processes are bimetallic systems composed of two different metals either as an alloy or in an intermetallic form. Dilution of a more active metal with a less active one significantly modifies the geometric and electronic structure of an active center, providing isolated active metal atoms at high dilutions (Hannagan et al., 2020). It was demonstrated previously that in addition to an enhanced chemical selectivity to an alkene, Pd site isolation by In in the intermetallic 1 wt.% Pd-In catalyst provides an enhanced selectivity to pairwise addition of $H_2$ to an alkyne (Burueva et al., 2018).

However, so far the vast majority of reported HET-PHIP experiments involving supported metal catalysts were performed with monometallic systems, whereas the potential advantages of bimetallic nanoparticles in this context were addressed in a very few studies. In this work, for the first time we directly compare monometallic and bimetallic Pd-based catalysts in the selective hydrogenation of propyne to propene with parahydrogen. The results clearly indicate that bimetallic Pd-Ag catalyst largely outperforms its monometallic counterparts not only in the overall activity and selectivity toward propene, but also in the pairwise selectivity of hydrogen addition as revealed by significantly larger NMR signal enhancements for propene. Furthermore, the choice of the dilution partner has a pronounced effect on the efficiency of bimetallic catalysts in HET-PHIP, as revealed by comparing Pd-Ag and Pd-In bimetallic catalysts.

# 2. Materials and methods

## 2.1. Catalyst preparation and characterization

Pd-Ag/Al$_2$O$_3$ catalyst was obtained via incipient wetness impregnation of Al$_2$O$_3$ (Sasol, specific surface area 56 m$^2$·g$^{-1}$) preliminarily calcined at 550 °C in flowing air for 3 h with an aqueous solution of Pd(NO$_3$)$_2$ and AgNO$_3$. To prepare this solution, 0.153 g of silver(I) nitrate (Merck, 204390-10G) was dissolved in 0.7059 g of 10 wt.% palladium(II) nitrate solution (Aldrich, 380040-50 ML). After that, 0.25 mL of distilled water was added. The resulting solution was used for

impregnation of 1.5 g of $Al_2O_3$. The product was dried overnight at room temperature in air and then reduced in 5 vol.% $H_2$/Ar flow (~ 100 mL/min) at 550 °C for 3 h. The temperature was increased from room temperature to 550 °C with a 3.5 °C/min ramp.

The reference catalyst samples ($Pd/Al_2O_3$ and $Ag/Al_2O_3$) were also prepared by incipient wetness impregnation with an aqueous solution of the corresponding nitrates. The impregnated samples were dried in air at room temperature overnight followed by the reduction in 5 vol.% $H_2$/Ar (~ 100 mL/min) flow at 550 °C for 3 h. The detailed preparation procedures of the catalysts used in this work are described elsewhere (Markov et al., 2016; Smirnova et al., 2019).

For preparation of the $Pd-In/Al_2O_3$ catalyst, an aqueous solution of binuclear acetate complex Pd $(OOCMe)_4In(OOCMe)$ was used as a precursor. The preparation procedure of this complex is described elsewhere (Stolarov et al., 2018). To obtain the impregnating solution, 0.164 g of $Pd(OOCMe)_4In(OOCMe)$ was dissolved in 4.75 mL of distilled water. After that, 1.45 g of $Al_2O_3$ (Sasol, specific surface area 56 $m^2 \cdot g^{-1}$) preliminarily calcined in flowing air (550 °C, 3 h) was impregnated by 0.95 mL of the solution followed by overnight drying at room temperature. The impregnation/drying procedure was repeated 5 times to overcome the insufficient solubility of the complex and achieve the required metal content with the "dry" impregnation method used. The resulting material was reduced at 600 °C for 3 h in flowing 5 vol.% $H_2$/Ar (~ 100 mL/min). The temperature was increased from room temperature to 600 °C with a 3.5 °C/min ramp.

The inductively coupled plasma atomic emission spectra (ICP-AES) were recorded on a Baird Plasma Spectrometer PS-6. According to ICP-AES data, $Pd-Ag/Al_2O_3$ catalyst sample contained 2 wt.% of Pd and 6 wt.% of Ag; $Pd-In/Al_2O_3$ catalyst contained 2 wt.% of Pd and 2 wt.% of In. The reference catalyst sample $Pd/Al_2O_3$ contained 2.5 wt.% of Pd; $Ag/Al_2O_3$ catalyst contained 6 wt.% of Ag. The Pd-Ag catalyst was extensively characterized in recent work by transmission electron microscopy (TEM), diffuse reflectance Fourier transform infrared spectroscopy of chemisorbed CO (DRIFTS-CO), temperature-programmed reduction with $H_2$ ($H_2$-TPR), and hydrogen temperature-programmed desorption ($H_2$-TPD) (Rassolov et al., 2020b). The structure of $Pd-In/Al_2O_3$ catalyst was studied in detail by TEM, CO-DRIFTS, and X-ray photoelectron spectroscopy (Burueva et al., 2018; Markov et al., 2019; Mashkovsky et al., 2018).

## 2.2. Catalytic activity tests

Commercially available hydrogen and propyne gases were used without additional purification. For catalytic tests, propyne was premixed with p-$H_2$-enriched hydrogen in the molar ratio of 1 : 4. Hydrogen gas was enriched with para-isomer up to 87.0-90.5% using Bruker parahydrogen generator BPHG-90. The catalyst (30 mg, density 0.67 $g \cdot cm^{-3}$) was placed in the middle of a stainless steel tubular reactor (1/4'' OD, 4.2 mm ID, 20 cm total length) between two plugs of fiberglass tissue. The bimetallic catalysts (Pd-Ag, Pd-In) as well as monometallic Ag catalyst were reduced in $H_2$ flow at 550 °C for 1 h before the catalytic tests. $Pd/Al_2O_3$ catalyst was reduced in $H_2$ flow at 300 °C for 1 h. The catalyst was cooled down to the desired reaction temperature without $H_2$ flow termination and the propyne/p-$H_2$ mixture was introduced to the catalyst. The reactor was positioned outside an NMR magnet and the substrate gas mixture was supplied to the reactor and then the

resulting mixture was supplied to the standard screw-cap 10-mm OD NMR tube (Merck, Z271969) placed inside NMR spectrometer for detection (ALTADENA experimental protocol, Pravica and Weitekamp, 1988) though a 1/16'' OD (1/32'' ID) PTFE capillary. In NMR tube the gas mixture was flowing from the bottom to the top and then to the vent through 1/4'' OD (5.8 mm ID) PTFE tubing connected with screw-ending of the NMR tube. All hydrogenation experiments were performed at ambient pressure (ca. 101 kPa). The reactor was heated with a tube furnace and the temperature was varied from 100 to 300 °C (in case of Pd-Ag catalyst) and to 500 °C (Pd-In catalyst) in 100 °C increments (the heating rate was 10 °C/min). The temperature was controlled with a K-type thermocouple placed immediately adjacent to the catalyst bed on the external side of the reactor. The reaction was conducted in a continuous flow regime with the reactor outflow continuously supplied to the probe of an NMR spectrometer and analyzed by $^1$H NMR. The gas flow rate was controlled using an Aalborg rotameter and varied stepwise from 1.3 to 3.8 mL/s. The gas flow was periodically interrupted in order to acquire stopped-flow $^1$H NMR spectra for evaluating conversion and selectivity. After the introduction of the substrate gas mixture to the catalyst and establishment of the steady-state regime, $^1$H NMR spectra were recorded on a 300 MHz Bruker AV NMR spectrometer using a $\pi/2$ rf pulse. A 10-mm BBO 300 MHz Bruker probehead was used.

The products of propyne hydrogenation are propene and propane; at high reaction temperatures ($\geq 400$ °C) propyne can isomerize to propadiene. The propyne conversion value ($X$) at a certain flow rate and temperature was calculated as the molar ratio of the reaction products (propene, propane, and propadiene) to the sum of products and unreacted propyne. The selectivity to propene ($S_{propene}$) was calculated as the molar ratio of propene to the sum of propene, propane, and propadiene. Both values were evaluated from $^1$H NMR spectra acquired in thermal equilibrium after a complete relaxation of hyperpolarization. The error of the quantitative analysis of gas-phase NMR spectra was estimated as 10%.

The activities of different catalysts in pairwise hydrogen addition were compared using the "apparent signal enhancement" values (SE), evaluated as the ratio between the integral of the enhanced NMR signal of propene CH- group and the integral of the corresponding signal for thermally polarized propene. Selectivity toward pairwise addition of $H_2$ is the estimated measure of the contribution of the pairwise $H_2$ addition to the overall mechanism of hydrogenation which is predominantly non-pairwise. It can be evaluated as the ratio of the "apparent" SE to the largest theoretically possible enhancement under conditions which ensure pairwise $H_2$ addition exclusively. For $^1$H polarization at 298 K, magnetic field of 7.1 T, and 90.5% fraction of p-$H_2$ the theoretical SE equals ~ 35880 (Bowers, 2007). The relaxation losses during the transfer of hyperpolarized propene to the NMR spectrometer for detection and the effect of signal suppression in the NMR spectra of continuously flowing gas mixture were not taken into account. So, the presented apparent SE values are lower estimates.

# 3. Results and Discussion

Here in this work we studied the catalytic behavior of different Pd-based bimetallic catalysts (Pd-Ag and Pd-In) in the selective gas-phase hydrogenation of propyne with parahydrogen.

First, the synergetic effect of Pd-Ag catalyst in HET-PHIP is addressed, then the behavior of this catalyst is compared to that of Pd-In, and after that factors affecting pairwise $H_2$ addition selectivity are discussed.

## 3.1.      Synergetic effect of Pd-Ag/Al$_2$O$_3$ catalyst in HET-PHIP

The catalytic activity of the monometallic reference catalysts (6 wt.% Ag/Al$_2$O$_3$ and 2 wt.% Pd/Al$_2$O$_3$) was explored first. It was found that no hydrogenation products were observed in the [1]H NMR spectra during the attempted propyne

hydrogenation over Ag/Al$_2$O$_3$ catalyst for the entire reaction temperature range, indicating that catalytic activity of Ag/Al$_2$O$_3$ is negligible under the experimental conditions. In contrast, the Pd/Al$_2$O$_3$ catalyst showed an excellent activity – even at a relatively low reaction temperature of 100 °C and the flow rate of 1.3 mL·s$^{-1}$ the propyne conversion reached 100%. However, the monometallic Pd/Al$_2$O$_3$ catalyst exhibited poor selectivity toward propene (Table 1). Slightly higher selectivity values at high gas mixture flow rates is explained by the lower contact times between the reacting gas and the catalyst. In

addition, a declining alkene selectivity at higher alkyne conversions is a known problem associated with unmodified Pd catalysts.

**Table 1: Hydrogenation of propyne with parahydrogen over Ag/Al$_2$O$_3$, Pd/Al$_2$O$_3$, and Pd-Ag/Al$_2$O$_3$ catalysts: propyne conversion (*X*) and selectivity to propene (*S$_{propene}$*) at different gas mixture flow rates and reaction temperature of 200 °C.**

| Catalyst | Flow rate, mL·s$^{-1}$ | *X*, % | S$_{propene}$, % |
|---|---|---|---|
| Ag | 1.3 | 0 | 0 |
|  | 3.8 | 0 | 0 |
| Pd | 1.3 | 100 | 45 |
|  | 3.8 | 92 | 71 |
| Pd-Ag | 1.3 | 100 | 75 |
|  | 3.8 | 78 | 86 |

Modification of Pd catalysts with a second metal, which is typically less active or even completely inactive in hydrogenation, leads to synergetic effect in catalytic activity – bimetallic Pd-based catalysts usually demonstrate enhanced selectivity and stability (Bond, 2005). Here we show that the introduction of Ag atoms inactive in propyne hydrogenation to Pd and formation of a bimetallic Pd-Ag catalyst (2 wt.% of Pd and 6 wt.% Ag) dramatically enhances its catalytic behavior. The Pd-Ag catalyst demonstrated a significantly higher selectivity toward propene with a comparable conversion in propyne

hydrogenation, e.g. at 100 °C and 1.3 mL·s$^{-1}$ both Pd and Pd-Ag catalysts showed 100% conversion, but Pd-Ag catalyst showed higher selectivity compared to monometallic Pd catalyst (75 vs. 45%).

        Such catalytic behavior is typical for bimetallic catalysts and can be associated with suppression of the formation of palladium hydride phase, which is a known hydrogen source responsible for the undesired unselective hydrogenation of alkynes and alkenes to alkanes (Armbrüster et al., 2012). Nevertheless, the enhanced catalytic performance of Pd-Ag catalyst

can be also explained by the fact that atomically dispersed Pd sites ($Pd_1$) isolated from each other by inactive Ag atoms can be achieved at high dilutions and stabilized during high-temperature pre-reduction in $H_2$ (400–550 °C) (Pei et al., 2015). The authors associate the higher selectivity of isolated $Pd_1$ sites over Pd-Ag catalyst with the decrease in the heat of adsorption of alkene compared to that on a monometallic Pd catalyst. The differentiation between the effects associated with the formation of isolated $Pd_1$ sites and the suppression of the formation of palladium hydrides for Pd-Ag catalysts with different Pd/Ag

ratio was thoroughly studied in previous works (Rassolov et al., 2020b, 2020a). It was found that an increase in the Ag content in the composition of bimetallic Pd-Ag nanoparticles hinders the formation of palladium hydrides, which is completely suppressed at a ratio of Ag/Pd $\geq$ 1. Analysis of the structural stability of isolated $Pd_1$ sites has shown that the stability of such sites can be provided by an increase in the Ag/Pd ratio to $\geq$ 2.

       The significant dilution of Pd atoms with Ag (Ag/Pd = 3) in this study ensures the formation of isolated $Pd_1$ sites,

which is confirmed by IR spectroscopy of adsorbed CO (Rassolov et al., 2020b). In an analogy with transition metal complexes in solution, the presence of isolated $Pd_1$ sites on the Pd-Ag catalyst surface can be expected to result in an enhanced activity of the bimetallic Pd-Ag catalyst in pairwise hydrogen addition. This expectation is indeed borne out by the experimental observations. Both monometallic Pd and bimetallic Pd-Ag catalysts were active in pairwise $H_2$ addition in propyne hydrogenation with parahydrogen (Figure 1). The enhanced NMR signals from CH- (signal # 4 in Figure 1) and

$CH_2$-protons (signals # 3 and # 5) of propene were observed.

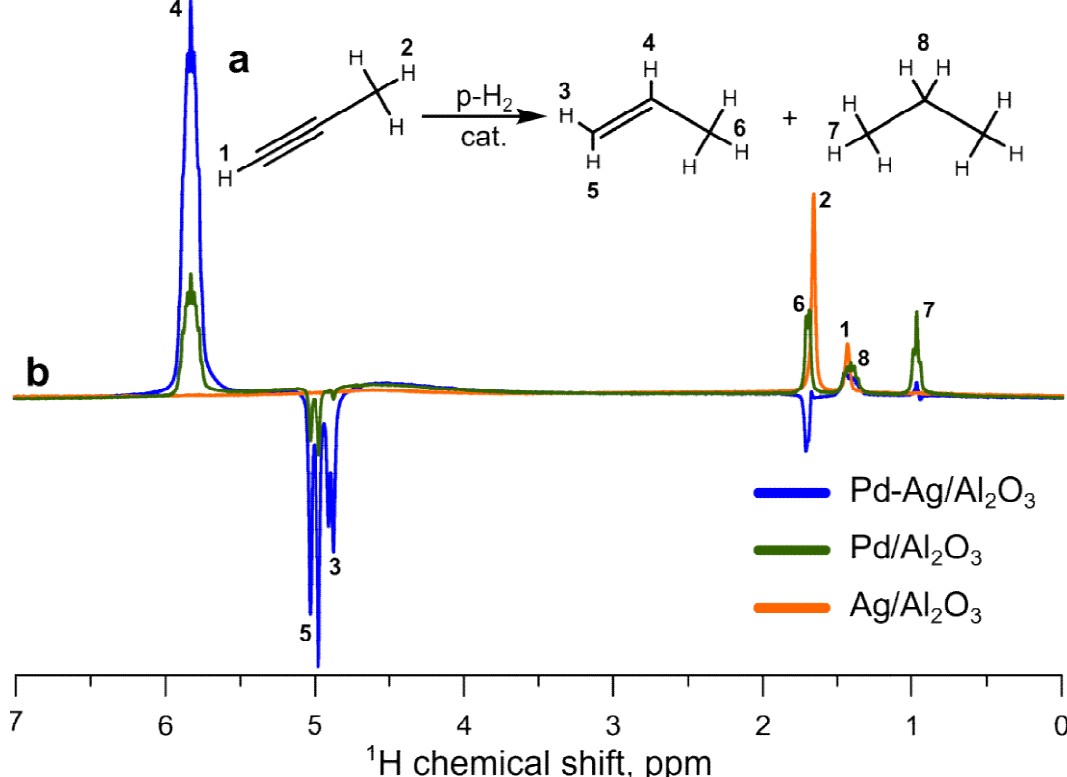

**Figure 1: a) Reaction scheme of propyne hydrogenation; b) $^1$H NMR ALTADENA spectra acquired during propyne hydrogenation with parahydrogen over Pd-Ag/Al$_2$O$_3$ (blue line), Pd/Al$_2$O$_3$ (green line), or Ag/Al$_2$O$_3$ catalyst (orange line). The reaction temperature was 200 °C, the total gas flow rate was 3.8 mL·s$^{-1}$. All spectra were acquired with 8 signal accumulations and are presented on the same vertical scale.**

The reference monometallic 2 wt.% Pd/Al$_2$O$_3$ catalyst demonstrated very low $^1$H NMR signal enhancement factors for CH-proton of propene, with SE values not exceeding 3 over the entire reaction temperature range. However, the dilution of Pd with Ag atoms allowed a ~4-fold increase in SE (up to 13) for the Pd-Ag catalyst (Table 2). The moderate activity of bimetallic Pd-Ag catalyst in pairwise H$_2$ addition may be associated with its tendency to the surface transformations – in an earlier work it was shown that on the surface of Pd-Ag catalyst a transformation of a part of monoatomic Pd$_1$ centers to multiatomic Pd$_n$ surface clusters takes place as a result of Pd atoms migration on the surface upon treatment with CO (Stakheev et al., 2018) due to adsorbate-induced segregation phenomenon. Most likely, analogous surface transformations are initiated by the strong adsorption of propyne. In the previous study it was found that such surface transformation almost didn't change catalyst selectivity in a liquid-phase hydrogenation process; however, our results clearly show that pairwise hydrogen addition is an extremely sensitive reaction and can be potentially used as a marker of the structural transformations of the catalyst surface.

**Table 2: Hydrogenation of propyne with parahydrogen over Ag/Al$_2$O$_3$, Pd/Al$_2$O$_3$ and Pd-Ag/Al$_2$O$_3$ catalysts: propyne conversion ($X$) and $^1$H NMR signal enhancement factors ($SE$) for the CH-proton of propene. The p-H$_2$ fraction was 90. 5%.**

| Catalyst | T, °C | Flow rate, mL·s$^{-1}$ | X, % | SE |
|---|---|---|---|---|
| Ag | 200 | 1.3 | 0 | – |
| | | 3.8 | 0 | – |
| Pd | 100 | 1.3 | 100 | - |
| | | 3.8 | 94 | 3 |
| | 200 | 1.3 | 100 | - |
| | | 3.8 | 92 | 3 |
| | 300 | 1.3 | 100 | - |
| | | 3.8 | 96 | 3 |
| Pd-Ag | 100 | 1.3 | 100 | – |
| | | 3.8 | 87 | 10 |
| | 200 | 1.3 | 100 | – |
| | | 3.8 | 78 | 13 |
| | 300 | 1.3 | 96 | – |
| | | 3.8 | 77 | 13 |

## 3.2. The effect of second metal M (Ag vs. In) on the activity of Pd-M/Al$_2$O$_3$ catalysts in HET-PHIP

As it was shown above, the stability of atomically dispersed Pd$_1$ sites on the surface may have a great impact on the efficiency of bimetallic catalysts in HET-PHIP. In order to differentiate the stability effect, Pd atoms were diluted with a different metal, In. The 2 wt.% Pd-In catalyst (Pd/In = 1) was also tested in propyne hydrogenation with parahydrogen. In these experiments, the slightly lower p-H$_2$ enrichment was used; the p-H$_2$ fraction was 87.0% (instead of 90.5% in case of Pd, Ag, and Pd-Ag catalytic activity tests). Therefore, for the sake of comparison, the apparent values of signal enhancement for Pd-In catalyst were extrapolated to 90.5% p-H$_2$ fraction (see column *SE corrected* in Table 3). The Pd-In catalyst clearly demonstrated a higher activity in pairwise hydrogen addition compared to the Pd-Ag catalyst, e.g. at 200 °C and 3.8 mL·s$^{-1}$ the signal enhancement was 88, i.e., much larger than for Pd-Ag (13). The comparison of hyperpolarized $^1$H NMR spectra acquired during propyne hydrogenation over Pd-Ag and Pd-In catalysts is presented in Figure 2; the spectra are shown on the same vertical scale. It is interesting to note that Pd-In catalyst demonstrated also a higher propene selectivity along with higher activity in HET-PHIP – at 200 °C and 1.3 mL·s$^{-1}$ the selectivity toward propene reached 98%, while for Pd-Ag it was 75%. The highest values of SE were observed at 400 °C – at 3.8 mL·s$^{-1}$ the signal enhancement was 113 for Pd-In.

**Table 3: Hydrogenation of propyne with parahydrogen over Pd-In/Al$_2$O$_3$ catalyst: propyne conversion (*X*), selectivity to propene (*S$_{propene}$*), and $^1$H NMR signal enhancement factors (*SE*) for the CH-proton of propene. The p-H$_2$ fraction was 87%. SE values extrapolated to 90.5% p-H$_2$ fraction are provided in the column *SE corrected*.**

| Catalyst | T, °C | Flow rate, mL·s$^{-1}$ | X, % | S$_{propene}$, % | SE | SE corrected* |
|---|---|---|---|---|---|---|
| Pd-In | 100 | 1.3 | 5 | 94 | 7 | 8 |
| | | 3.8 | 4 | 93 | 59 | 62 |
| | 200 | 1.3 | 93 | 98 | 4 | 4 |
| | | 3.8 | 63 | 97 | 83 | 88 |
| | 300 | 1.3 | 89 | 98 | 4 | 4 |
| | | 3.8 | 45 | 96 | 89 | 95 |
| | 400 | 1.3 | 62 | 94 | 3 | 3 |
| | | 3.8 | 19 | 91 | **107** | **113** |
| | 500 | 1.3 | 24 | 48 | 7 | 8 |
| | | 3.8 | 12 | 47 | 86 | 91 |

\* SE values corrected to the 90.5% parahydrogen fraction

## 3.3. Factors affecting pairwise H$_2$ addition selectivity

The NMR signal enhancements observed in this study are rather moderate, and higher values have been achieved in HET-
PHIP experiments with bimetallic catalysts in the past. For instance, higher SE values were observed in the previous work
for 1 wt.% Pd-In/Al$_2$O$_3$ catalyst (Burueva et al., 2018), which is likely associated with a different metal loading. Also,
substantially higher enhancements were reported for Pt-Sn intermetallic nanoparticles confined within mesoporous silica (Du
et al., 2020). However, it should be stressed that the activity of the Pd, Pd-Ag, and Pd-In catalysts is compared here using the
"apparent signal enhancement" values. The catalytic tests were carried out under ALTADENA experimental protocol
(Pravica and Weitekamp, 1988), implying that hydrogenation reaction proceeds outside a NMR spectrometer, with
subsequent delivery of catalytic reactor outflow over a substantial distance to the probe of the NMR spectrometer for
detection. Hence, the apparent signal enhancements evaluated for Pd-based catalysts presented in Table 2 and Table 3 are
lower estimates of the actual values of initial enhancements, underestimated (and possibly significantly) because
hyperpolarization relaxation dramatically reduces the intensities of enhanced $^1$H NMR signals of propene during the transfer.
The accurate analysis of processes leading to polarization losses in the ALTADENA experiment (non-adiabaticity of
magnetic field change during the transfer of hyperpolarized product from the Earth's magnetic field, the relaxation losses in
both high and low magnetic fields, etc.) performed previously (Barskiy et al., 2015; Burueva et al., 2018) indicates that the
apparent signal enhancement factors are significantly reduced due to the abovementioned causes – by one order of
magnitude or possibly more. Also, during the delivery of hyperpolarized products to the NMR probe the polarization is
redistributed in the Earth's magnetic field, leading to the observation of polarized signal of the CH$_3$- group of propene
(negative signal #6 in blue spectrum, which was acquired while Pd-Ag catalyst was used), which cannot be polarized directly
by pairwise hydrogen addition to propyne. In addition, Figure 1 clearly shows that for both Pd and Pd-Ag catalysts the broad

signal at 4.5 ppm (from orthohydrogen) emerges in the $^1$H NMR spectra, meaning that ortho-para conversion of $H_2$ takes place on these catalysts. The concrete mechanism of this process and its impact on pairwise hydrogen addition is unclear at present, and further detailed investigation is required. Overall, these and other factors lead to the underestimation of the true selectivity to pairwise $H_2$ addition over the catalysts studied, which can be rahter significant. At the same time, while minimization of relaxation losses is very important for applications of HET-PHIP, the primary objective of this work is the exploration of how the nature of a catalyst affects its selectivity toward pairwise $H_2$ addition to a substrate.

The Pd-In bimetallic catalyst is shown to provide a significantly more pronounced PHIP effect compared to the Pd-Ag system. One potential reason for the inferior behavior of the Pd-Ag catalyst is already mentioned earlier, namely the restructuring of the surface under reactive conditions in the presence of adsorbates such as propyne and $H_2$. In contrast to Pd-Ag solid solution alloy, Pd-In surface structure is more stable owing to the high formation enthalpy of the Pd-In intermetallic compound, which prevents surface segregation even in the presence of adsorbates.

However, there are potentially other contributing factors. While the Ag/Al$_2$O$_3$ catalyst was completely inactive in propyne hydrogenation, this does not mean that hydrogenation cannot proceed on the Ag atoms of the Pd-Ag catalyst. Indeed, the primary reason of the inactivity of metals such as Ag and Au in catalytic hydrogenations is known to be their inability to efficiently activate $H_2$ due to a large activation energy barrier for dissociative $H_2$ chemisorption on these metals. In bimetallic catalysts containing Ag or Au in combination with a platinum group metal such as Pd, a new reaction channel becomes available. It involves dissociative and often essentially barrierless chemisorption of $H_2$ on a Pd atom/cluster followed by spillover of the resulting hydrogen atoms to the other metal (e.g., Ag or Au) which can then efficiently incorporate atomic hydrogen into the hydrogenation product (Hannagan et al., 2020). As hydrogen spillover tends to randomize hydrogen atoms on the catalyst surface, such hydrogenation mechanism cannot achieve pairwise $H_2$ addition to a substrate. This may be an additional reason why Pd-Ag combination is inferior to Pd-In in the production of HET-PHIP. Furthermore, in addition to the ensemble effects associated with the isolation of individual Pd atoms as catalytically active centers when Pd is diluted with a less active metal, the electronic structure of Pd is altered upon dilution, and this alteration will significantly depend on the electronic (e.g., electron donating or withdrawing) properties of the added metal.

Importantly, in the current study we show that Pd-containing bimetallic catalysts significantly outperform monometallic Pd catalysts in terms of the achievable NMR signal enhancements in heterogeneous hydrogenations of unsaturated compounds with parahydrogen, while at the same time maintain very high activity as well as hydrogenation selectivity toward semihydrogenation of alkynes. This is in contrast to the behavior of Pt-based catalysts reported earlier (Du et al., 2020), which demonstrated a significant decline in activity upon progressive dilution with Sn, combined with usually lower selectivity of Pt in the hydrogenation of alkynes to alkenes compared to Pd. This is in agreement with the fact that Pd-based catalysts are preferred in industrial hydrogenations to achieve high reaction yields and product selectivities. Furthermore, the results presented above clearly show that the choice of the dilution metal for bimetallic catalysts has a pronounced effect on their efficiency in HET-PHIP. This efficiency is governed by multiple factors, including possible catalyst surface restructuring under reactive conditions, electronic effects exerted by the secondary metal on the catalytically

active component, and involvement of additional reaction pathways such as "metal-ligand cooperation" when both metals are involved in the hydrogenation event. Further detailed studies are required to establish which factors are the most relevant, which will eventually provide guidance to achieving the ultimate NMR signal enhancements in HET-PHIP experiments.

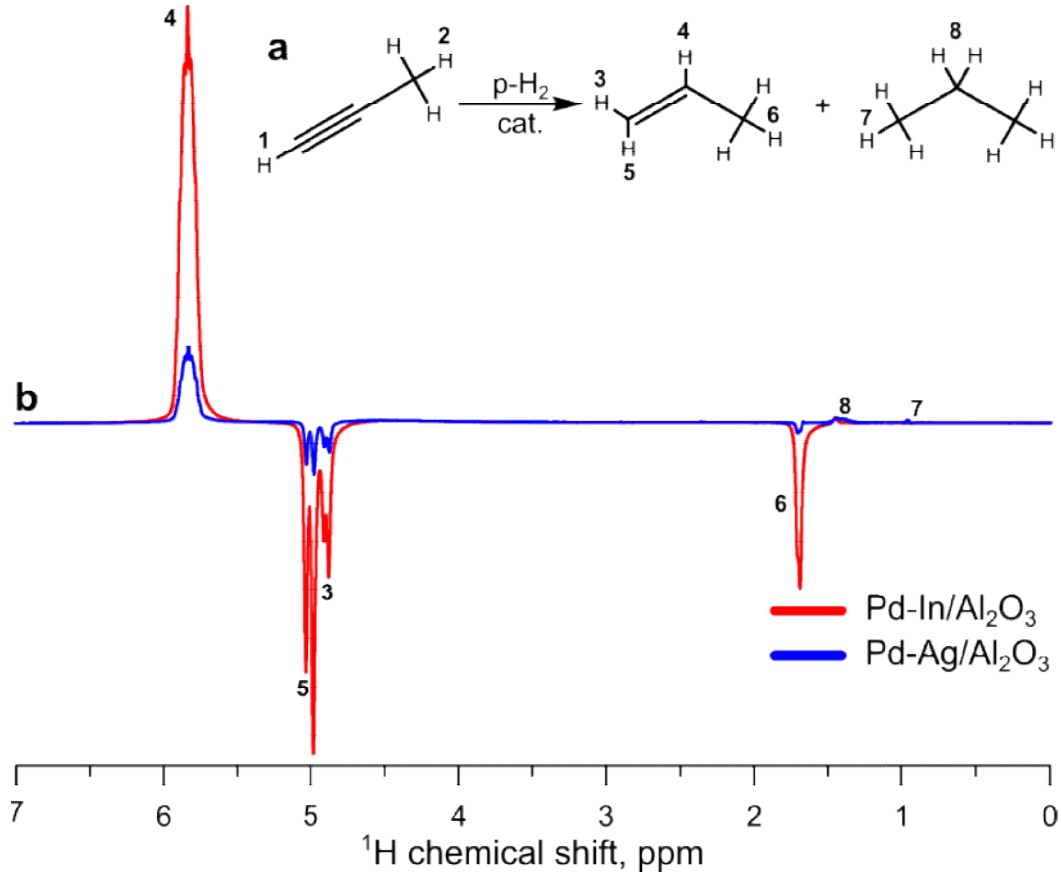

**Figure 2: a) Reaction scheme of propyne hydrogenation; b) $^1$H NMR ALTADENA spectra acquired during propyne hydrogenation with parahydrogen over Pd-In/Al$_2$O$_3$ (red line) or Pd-Ag/Al$_2$O$_3$ (blue line; the same spectrum as in Fig. 1) catalysts. The reaction temperature was 200 °C, the total gas flow rate was 3.8 mL·s$^{-1}$. All spectra were acquired with 8 signal accumulations and are presented on the same vertical scale.**

# 4. Conclusions

Further progress in the development of parahydrogen-induced polarization in heterogeneous hydrogenation reactions is beneficial for multiple fundamental and practical applications. These include facile production of catalyst-free hyperpolarized liquids and gases (including their continuous production) for numerous purposes including in vivo studies, a possibility to gain deeper insight into the detailed mechanisms of industrially important catalytic processes, enrichment of nuclear spin isomers of polyatomic gases, and more. All this requires novel advanced strategies for increasing efficiency of heterogeneous catalysts in the pairwise H$_2$ addition to unsaturated substrates. One such strategy addressed here is the use of

heterogeneous bimetallic catalysts in which a more active metal is diluted by a less active one, thereby providing the structure reminiscent of single-metal-atom catalytic centers of homogeneous transition metal complexes. Indeed, it is demonstrated that the high hydrogenation activity of Pd metal is largely retained upon dilution in the Pd-Ag and Pd-In catalysts, while at the same time the selectivity toward pairwise $H_2$ addition to propyne is measurably enhanced compared to the monometallic Pd system. Furthermore, In is shown to be a better choice compared to Ag as the secondary metal in bimetallic Pd-M compositions, which is likely associated with differences in the tendency toward surface restructuring, the potential involvement of additional non-pairwise reaction channels, and differences in ensemble effects and electronic structures of active centers in the two compositions. While the exact factors that govern pairwise efficiency in bimetallic structures are yet to be fully established, the results demonstrate that further search for more efficient bimetallic catalysts is warranted in order to advance this field of research.

**Data availability.** NMR data relevant to this publication are available at https://doi.org/10.5281/zenodo.4436159 (Burueva et al., 2021).

**Author contributions.** IVK designed the research. AYS prepared the catalysts. DBB performed PHIP experiments. IVK and DBB discussed the results and wrote the paper. The paper was revised by all authors.

**Competing interests.** The authors declare that they have no conflict of interest.

**Financial support.** IVK and DBB acknowledge funding from the Ministry of Science and Higher Education of the Russian Federation (grant # 075-15-2020-779).

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
