# Peer review of "Pd-based bimetallic catalysts for HET-PHIP"

_Magnetic Resonance, 2021_

## Referee Comment (RC1)

1. Does the paper address relevant scientific questions within the scope of MR?

   The authors worked on HET-PHIP which is related to the field of NMR and hyperpolarization and fits the scope of MR.

2. Does the paper present novel concepts, ideas, tools, or data? All submitted papers are assumed to report on new observations and/or new theory; there is no need to draw attention to the novelty in title, abstract, or conclusions.

   On the one hand they tested a novel type of bimetallic heterogenous catalyst based on Pd-Ag or Pd-In. On the other hand, the work is embedded in a series of Pd type catalysts these authors used for the same reaction several times. The novelty compared to their former works should be highlighted more in detail.

3. Are substantial conclusions reached?

   The authors achieved further progress in the field of PHIP employing heterogenous catalyst systems. It is shown that a second metal such as Ag or In acts on the selectivity of the Pd catalyst and let achieve it more selectivity towards pairwise hydrogen transfer which is required to produce PHIP.

4. Are the results sufficient to support the interpretations and conclusions?

   In part they are sufficient but a deeper characterization of the system would be helpful. Did the authors obtain molecular bimetallic complexes or did they obtain metallic Pd nanoparticles that are doped with In or Ag to reach the selectivity? When talking about selectivity, what are the byproducts of this reaction?

   Is the description of experiments and calculations sufficiently complete and precise to allow their reproduction by fellow scientists with reasonable effort? Detailed technical and graphical explanations and documentation of limited file size can be provided as supporting information. Access to raw data, processed spectra, and other experimental data must be provided by depositing in a publicly accessible repository or archive as far as practically feasible, and the DOI provided in the article. Hardware developments need to be documented by photos or equivalent drawings (blueprints with precise dimensions if possible). New software must be accompanied by user instructions. New software should be open source and access to it provided through a software repository if possible.

   In part the experiments could be reproduced but I miss more details for the catalyst preparation. Especially amounts of precursors etc. used in the synthesis are missing. Furthermore, details on the catalytic tests are not provided. How did they perform the PHIP experiments? Such details should be provided either in the experimental section or in the ESI.

5. Are numerical data accompanied by error estimates with a description of the methods used to obtain these estimates?

   Can the authors provide errors of their calculated enhancement factors?

6. Do the authors give proper credit to related work and clearly indicate their own new/original contribution?

In part. The authors should consider additional citations on HET-PHIP in the introduction

[1]     U. Obenaus, S. Lang, R. Himmelmann, M. Hunger, J. Phys. Chem. C 2017, 121, 9953-9962.

[2]     T. Gutmann, T. Ratajczyk, Y. Xu, H. Breitzke, A. Gruenberg, S. Dillenberger, U. Bommerich, T. Trantzschel, J. Bernarding, G. Buntkowsky, Solid State Nuclear Magnetic Resonance 2010, 38, 90-96.

[3]     A. M. Balu, S. B. Duckett, R. Luque, Dalton Transactions 2009, 5074-5076.

Although these works do not name the technique HET-PHIP they show PHIP with heterogenous catalysts.

In the introduction the authors talked about DNP, but do not specify. When they talk about DNP in general also citations of recent reviews should be included.

[1]     A. G. M. Rankin, J. Trebosc, F. Pourpoint, J. P. Amoureux, O. Lafon, Solid State Nuclear Magnetic Resonance 2019, 101, 116-143.

[2]     A. S. L. Thankamony, J. J. Wittmann, M. Kaushik, B. Corzilius, Progress in Nuclear Magnetic Resonance Spectroscopy 2017, 102, 120-195.

[3]     U. Akbey, W. T. Franks, A. Linden, M. Orwick-Rydmark, S. Lange, H. Oschkinat, in Hyperpolarization Methods in NMR Spectroscopy, Vol. 338 (Ed.: L. T. Kuhn), 2013, pp. 181-228.

Line 66-67 "So far, however, most heterogeneous catalysts demonstrated a limited efficiency in the pairwise hydrogen addition, or in some cases the low yields of the desired product, or both (Kovtunov et al., 2013, 2016, 2020a)"

Here also the works of Duckett and of Buntkowsky and co-workers should be considered.

7. Does the title clearly reflect the contents of the paper?

Yes

8. Does the abstract provide a concise and complete summary?

Yes

9. Is the overall presentation well-structured and clear?

Yes

10. Is the language fluent and precise?

Yes, but few typos have to be corrected:

Line 199: should be "experimental"

Line 204: should be "hyperpolarized"

11. Should any parts of the paper (text, formulae, figures, tables) be clarified, reduced, combined, or eliminated?

In principle table 2 and table 3 can be combined in one table.

12. Are the number and quality of references appropriate?

In part. See comment 6

13. Is the amount and quality of the supporting information and supplementary material appropriate?

There is no ESI available

---

## Author Comment (AC1)

Replies to the comments of Anonymous Referee # 1.

1. Does the paper address relevant scientific questions within the scope of MR?

The authors worked on HET-PHIP which is related to the field of NMR and hyperpolarization and fits the scope of MR.

2. Does the paper present novel concepts, ideas, tools, or data? All submitted papers are assumed to report on new observations and/or new theory; there is no need to draw attention to the novelty in title, abstract, or conclusions.

On the one hand they tested a novel type of bimetallic heterogenous catalyst based on Pd-Ag or Pd-In. On the other hand, the work is embedded in a series of Pd type catalysts these authors used for the same reaction several times. The novelty compared to their former works should be highlighted more in detail.

**Authors' response:**

We have changed the first sentence of the last paragraph in the introduction to better highlight the novelty of this work. This part now reads:

"However, so far the vast majority of reported HET-PHIP experiments involving supported metal catalysts were performed with monometallic systems, whereas the potential advantages of bimetallic nanoparticles in this context were addressed in only a very few studies. In this work, for the first time we directly compare monometallic and bimetallic Pd-based catalysts in the selective hydrogenation of propyne to propylene with parahydrogen."

3. Are substantial conclusions reached?

The authors achieved further progress in the field of PHIP employing heterogenous catalyst systems. It is shown that a second metal such as Ag or In acts on the selectivity of the Pd catalyst and let achieve it more selectivity towards pairwise hydrogen transfer which is required to produce PHIP.

4. Are the results sufficient to support the interpretations and conclusions?

In part they are sufficient but a deeper characterization of the system would be helpful. Did the authors obtain molecular bimetallic complexes or did they obtain metallic Pd nanoparticles that are doped with In or Ag to reach the selectivity?

**Authors' response:**

For the preparation of $Pd/Al_2O_3$, $Ag/Al_2O_3$ and $Pd-Ag/Al_2O_3$, aqueous solutions of $Pd(NO_3)_2$, $AgNO_3$ or both were used, respectively. $Pd-In/Al_2O_3$ was prepared using an aqueous solution of binuclear acetate complex $Pd(OOCMe)_4In(OOCMe)$.

We have added the detailed description of the preparation procedures of catalysts to the revised manuscript.

For convenience, it is repeated here:

1) Pd-Ag/Al2O3 catalyst was obtained via incipient wetness impregnation of parent Al2O3 (Sasol, specific surface area 56 m2•g-1) preliminarily calcined at 550 ºC in flowing air for 3 h with an aqueous solution of Pd(NO3)2 and AgNO3.To prepare this solution, 0.153 g of silver (I) nitrate (Merck, 204390-10G) was dissolved in 0.7059 g of 10 wt.% palladium (II) nitrate solution (Aldrich, 380040-50ML). After that, 0.25 mL of distilled water was added. The resulting solution was used for impregnation of 1.5 g of Al2O3. The product was dried overnight at room temperature in air and then reduced in 5 vol.% H2/Ar flow (~100 mL/min) at 550 ºC for 3 h. The temperature was increased from room temperature to 550 ºC with a 3.5 ºC/min ramp.

2) For preparation of Pd-In/Al2O3 catalyst, an aqueous solution of binuclear acetate complex Pd(OOCMe)4In(OOCMe) was used as a precursor. This complex was synthesized as follows: the reaction between Pd3(OOCMe)6 and In(OOCMe)3 in glacial acetic acid results in the crystalline complex Pd(OOCMe)4In-(OOCMe)•HOOCMe as the crystal solvate with a 49% yield based on Pd. Its subsequent recrystallization from benzene produced the solvent-free complex Pd(OOCMe)4In(OOCMe) with the yield of 36% based on Pd. More details on its preparation and characterization can be found elsewhere [R1].

3) To obtain the impregnating solution, 0.164 g of Pd(OOCMe)4In(OOCMe) was dissolved in 4.75 mL of distilled water. After that, 1.45 g of Al2O3 (Sasol, specific surface area 56 m2/g) preliminarily calcined in flowing air (550 ºC, 3 h) was impregnated by 0.95 mL of the solution followed by overnight drying at room temperature. The impregnation/drying procedure was repeated 5 times to overcome insufficient solubility of the complex and achieve the required metal content with the "dry" impregnation method used. The resulting material was reduced at 600 ºC for 3 h in flowing 5 vol.% H2/Ar (~100 mL/min). The temperature was increased from room temperature to 600 ºC with a 3.5 ºC/min ramp.

The structure of Pd-Ag/$Al_2O_3$ was recently characterized in detail by TEM, CO-DRIFTS, $H_2$-TPR, and $H_2$-TPD [R2]. The structure of Pd-In/$Al_2O_3$ catalyst was studied in detail by TEM, CO-DRIFTS, and XPS [R3-R5].

This information on catalysts characterization was added to the revised manuscript.

For convenience, the main findings of the catalysts characterization are repeated here.

The morphology of Pd-Ag nanoparticles was studied by TEM showing spherical PdAg nanoparticles with a relatively narrow unimodal distribution with a maximum at 6–8 nm. The formation of single-atom $Pd_1$ sites on the surface of supported Pd–Ag bimetallic nanoparticles was proved by infrared spectroscopy of adsorbed CO by disappearance of the signal corresponding to bridged and hollow-bonded CO adsorption and an increase in the intensity of the linearly adsorbed CO band. Additionally it was demonstrated that the structure of $Pd_1$ single-atom sites is highly stable even under conditions of CO-induced segregation. The $H_2$-TPR data on PdAg particles clearly show that the reduction of both the Pd and Ag components of bimetallic catalysts occurs at temperatures below 150°C. It should be also noted that the characteristic feature corresponding to PdH decomposition is absent indicating the suppression of the PdH formation in the PdAg alloyed nanoparticles. This data agrees well with that of $H_2$-TPD analysis.

TEM data demonstrate that Pd-In/$Al_2O_3$ catalyst contains nearly spherical cuboctahedral nanoparticles [R4]. The particle size distribution is relatively narrow between 2.5 and 6 nm centered at 4.5 nm. Formation of $Pd_1$ single-atoms isolated by In ones on the surface of the Pd.In intermetallic compound was confirmed via CO-DRIFTS by the absence of absorption bands below 2000 $cm^{-1}$ [R4,R5]. Such structure ensures the formation of isolated active $Pd_1$ sites on the catalyst surface and explains the absence of multi-point adsorption of CO, which requires the presence of several neighboring palladium atoms. Furthermore, the formation of PdIn intermetallic nanoparticles with $Pd_1$ isolated sites was shown by XPS [R1,R4]. The characteristic shift of the Pd 3d5/2 and In 3d5/2 lines was observed from 334.9 to 335.9 eV and from 445.2 to 443.4 eV, respectively. This fact indicates the redistribution of the electron density and formation of PdIn intermetallics. This data agrees well with XRD data [R5]. Thus analysis of XRD patterns reveals the formation of $Pd_1In_1$ intermetallic phase of cubic structure of the CsCl type, space group Pm-3m, a = 3.23(4) Å (3.246 Å for PdIn 59473-ICSD).

**References**

[R1] I.P. Stolarov, I.A. Yakushev, A.V. Churakov, N.V. Cherkashina, N.S. Smirnova, E.V. Khramov, Y.V. Zubavichus, V.N. Khrustalev, A.A. Markov, A.P. Klyagina, A.B. Kornev, V.M. Martynenko, A.E. Gekhman, M.N. Vargaftik, I.I. Moiseev, Heterometallic Palladium(II)−Indium(III) and −Gallium(III) Acetate-Bridged Complexes: Synthesis, Structure, and Catalytic Performance in Homogeneous Alkyne and Alkene Hydrogenation, Inorg. Chem. 2018, 57, 11482−11491, DOI: 10.1021/acs.inorgchem.8b01313

[R2] A. V. Rassolov, G. O. Bragina, G. N. Baeva, N. S. Smirnova, A. V. Kazakov, I. S. Mashkovsky, A. V. Bukhtiyarov, Ya. V. Zubavichus, A. Yu. Stakheev, Formation of Isolated Single-Atom $Pd_1$ Sites on the Surface of Pd–Ag/$Al_2O_3$ Bimetallic Catalysts, Kinet. Catal. 2020, 61 (5), DOI: 10.1134/S0023158420050080

[R3] I.S. Mashkovsky, N.S. Smirnova, P.V. Markov, G.N. Baeva, G.O. Bragina, A.V. Bukhtiyarov, I.P. Prosvirin, A.Yu. Stakheev, Tuning the surface structure and catalytic performance of PdIn/$Al_2O_3$ in selective liquid-phase hydrogenation by mild oxidative-reductive treatments, Mendeleev Commun., 2018, 28, 603-605; doi:10.1016/j.mencom.2018.11.013

[R4] P.V. Markov, A.V. Bukhtiyarov, I.S. Mashkovsky, N.S. Smirnova, I.P. Prosvirin, Z.S. Vinokurov, M.A. Panafidin, G.N. Baeva, Ya.V. Zubaichus, V.I. Bukhtiyarov, A.Yu. Stakheev, PdIn/$Al_2O_3$ Intermetallic Catalyst: Structure and Catalytic Characteristics in Selective Hydrogenation of Acetylene, Kinet. Catal., 2019, 60 (6), 842–850; doi: 10.1134/S0023158419060065

[R5] D.B. Burueva, K.V. Kovtunov, A.V. Bukhtiyarov, D.A. Barskiy, I.P. Prosvirin, I.S. Mashkovsky, G.N. Baeva, V.I. Bukhtiyarov, A.Yu. Stakheev, I.V. Koptyug, Selective single-site Pd-In hydrogenation catalyst for production of enhanced magnetic resonance signals using parahydrogen, Chem. Eur. J. 2018, 24, 2547-2553, DOI: 10.1002/chem.201705644

When talking about selectivity, what are the byproducts of this reaction?

**Authors' response:**

The term "selectivity' is used in two different contexts in this work: i) chemical selectivity toward propene in the hydrogenation of propyne; ii) selectivity toward pairwise $H_2$ addition to a substrate. We believe that this dual use of the term does not create any confusion, as its meaning is explained each time the term is used.

Generally, the only hydrogenation products of propyne are propene and propane. However, at relatively high temperatures ($\geq 400$ °C) the side reaction can occur – propyne can isomerize to propadiene. The decrease in selectivity to propene at 500 °C for Pd-In catalyst is associated with this side reaction. Therefore, the selectivity toward propene is evaluated as $S_{propene}= \dfrac{I_{propene}}{I_{propene}+I_{propane}+I_{propadiene}}$, where $I_{product}$ is the NMR signal intensity of the corresponding product (propene, propane, or propadiene) normalized by the number of protons contributing to that NMR signal.

Selectivity toward pairwise addition of $H_2$ is the estimated measure of the contribution of the pairwise $H_2$ addition to the overall mechanism of hydrogenation which is predominantly non-pairwise. It is evaluated as the ratio of the observed NMR signal enhancement to the largest theoretically possible enhancement under conditions which ensure that $H_2$ addition is exclusively pairwise.

This information was added in the revised manuscript.

Is the description of experiments and calculations sufficiently complete and precise to allow their reproduction by fellow scientists with reasonable effort? Detailed technical and graphical explanations and documentation of limited file size can be provided as supporting information. Access to raw data, processed spectra, and other experimental data must be provided by depositing in a publicly accessible repository or archive as far as practically feasible, and the

DOI provided in the article. Hardware developments need to be documented by photos or equivalent drawings (blueprints with precise dimensions if possible). New software must be accompanied by user instructions. New software should be open source and access to it provided through a software repository if possible.

In part the experiments could be reproduced but I miss more details for the catalyst preparation. Especially amounts of precursors etc. used in the synthesis are missing.

**Authors' response:**

The details about preparation and characterization of the catalysts were added to the revised manuscript and are also provided above.

Furthermore, details on the catalytic tests are not provided. How did they perform the PHIP experiments? Such details should be provided either in the experimental section or in the ESI.

**Authors' response:**

Following reviewer's suggestion, we have added in Section 2.2 of the revised manuscript the detailed description of how PHIP experiments were performed.

5. Are numerical data accompanied by error estimates with a description of the methods used to obtain these estimates?

Can the authors provide errors of their calculated enhancement factors?

**Authors' response:**

The uncertainty in the quantitative analysis of gas-phase NMR spectra was estimated as 10 %. This information was added in the recised manuscript.

6. Do the authors give proper credit to related work and clearly indicate their own new/original contribution?

In part. The authors should consider additional citations on HET-PHIP in the introduction

[1] U. Obenaus, S. Lang, R. Himmelmann, M. Hunger, J. Phys. Chem. C 2017, 121, 9953-9962.

[2] T. Gutmann, T. Ratajczyk, Y. Xu, H. Breitzke, A. Gruenberg, S. Dillenberger, U. Bommerich, T. Trantzschel, J. Bernarding, G. Buntkowsky, Solid State Nuclear Magnetic Resonance 2010, 38, 90-96.

[3] A. M. Balu, S. B. Duckett, R. Luque, Dalton Transactions 2009, 5074-5076.

Although these works do not name the technique HET-PHIP they show PHIP with heterogenous catalysts.

**Authors' response:**

It was an oversight on our part not to cite the work done by others in the introduction.

We have added several relevant references.

In the introduction the authors talked about DNP, but do not specify. When they talk about DNP in general also citations of recent reviews should be included.

[1] A. G. M. Rankin, J. Trebosc, F. Pourpoint, J. P. Amoureux, O. Lafon, Solid State Nuclear Magnetic Resonance 2019, 101, 116-143.

[2] A. S. L. Thankamony, J. J. Wittmann, M. Kaushik, B. Corzilius, Progress in Nuclear Magnetic Resonance Spectroscopy 2017, 102, 120-195.

[3] U. Akbey, W. T. Franks, A. Linden, M. Orwick-Rydmark, S. Lange, H. Oschkinat, in Hyperpolarization Methods in NMR Spectroscopy, Vol. 338 (Ed.: L. T. Kuhn), 2013, pp. 181-228.

**Authors' response:**

We believe the five references that we already have for the DNP technique, covering both MAS DNP and dissolution DNP, to be representative and sufficient, especially given that DNP is not considered and/or used in our work. Besides, at least two of the (solid-state) DNP references suggested by the reviewer are cited in the DNP references provided in our paper.

Line 66-67 "So far, however, most heterogeneous catalysts demonstrated a limited efficiency in the pairwise hydrogen addition, or in some cases the low yields of the desired product, or both (Kovtunov et al., 2013, 2016, 2020a)"

Here also the works of Duckett and of Buntkowsky and co-workers should be considered.

**Authors' response:**

We have added the relevant references.

7. Does the title clearly reflect the contents of the paper?

Yes

8. Does the abstract provide a concise and complete summary?

Yes

9. Is the overall presentation well-structured and clear?

Yes

10. Is the language fluent and precise?

Yes, but few typos have to be corrected:

Line 199: should be "experimental"

Line 204: should be "hyperpolarized"

**Authors' response:**

The typos have been corrected and the text spell-checked and proofread once again,

11. Should any parts of the paper (text, formulae, figures, tables) be clarified, reduced, combined, or eliminated?

In principle table 2 and table 3 can be combined in one table.

**Authors' response:**

This is certainly possible, but we do not see any real advantage in doing so.

12. Are the number and quality of references appropriate?

In part. See comment 6

**Authors' response:**

This has been taken care of (see above).

13. Is the amount and quality of the supporting information and supplementary material appropriate?

There is no ESI available

---

## Author Comment (AC2)

Replies to the comments of Anonymous Referee #2.

The authors present PHIP experiments in the gas phase of propyne employing heterogeneous bimetallic catalysts. The study provides useful information for improving HET-PHIP experiments but should be revised with respect to the following specific comments:

1) Line 97: "Pd-Ag/Al$_2$O$_3$ catalyst sample contained 2 wt.% of Pd and 6 wt.% of Ag; Pd-In/Al$_2$O$_3$ catalyst contained 2 wt.% of Pd and 2 wt.% of In". This result in different dilutions of Pd in the other (less catalytically active) metal. Therefore, the amount of catalytically active Pd$_1$ sites should be different in the two bimetallic catalysts? How does this impact on the comparability of the results for the Pd-Ag and the Pd-In catalyst?

**Authors' response:**

There is no doubt that the total number of catalytically active centers (e.g., Pd$_1$ sites) impacts the performance of a catalyst, and in particular the conversion of reactants into products. The conventional practice in catalysis is the normalization of conversion with respect to the number of e.g., surface metal atoms (or active centers). This normalization yields the turnover frequency (TOF) of a catalyst which characterizes the number of product molecules produced by one catalytic center per unit time.

However, in PHIP experiments we are dealing with an additional factor which is not usually relevant in fundamental and industrial catalysis, namely the selectivity of a catalyst toward pairwise H$_2$ addition to a substrate. Evaluation of pairwise selectivity and maximizing it by designing an appropriate catalyst is the primary objective of this study. Pairwise selectivity is evaluated from SE value which is the ratio of the polarized and the thermal NMR signals of the product molecule. It is thus already normalized with respect to the amount of product produced (conversion), i.e., is essentially calculated per one active site.

While we fully agree that the overall conversion is highly important in PHIP experiments for achieving the maximum possible NMR signal intensity, our current objective is to find the best catalyst in terms of reaction mechanism which sustains pairwise H$_2$ addition before attempting to maximize the overall yield of a hyperpolarized product.

2) Very few experimental details are presented in the entire manuscript. Reviewer 1 already asked for more details on the synthesis and analysis of the bimetallic catalysts, which was sufficiently answered by the authors in the revised version of the manuscript. This reviewer is more concerned with the understanding of the NMR experiments and the reaction control of the hydrogenation experiments, which are both not sufficiently described in the experimental section.

**Authors' response:**

Following reviewer's suggestion, we have added in Section 2.2 of the revised manuscript the detailed description of NMR experiments and the reaction control of the hydrogenation experiments.

*The first paragraph of section 2.2 now reads:*
"Commercially available hydrogen and propyne gases were used without additional purification. For catalytic tests, propyne was premixed with p-$H_2$-enriched hydrogen in the molar ratio of 1 : 4. Hydrogen gas was enriched with para-isomer up to 87.0-90.5% using Bruker parahydrogen generator BPHG-90. The catalyst (30 mg, density 0.67 g·$cm^{-3}$) was placed in a stainless steel tubular reactor (1/4'' OD, 4.2 mm ID, 20 cm total length) between two plugs of fiberglass tissue. The bimetallic catalysts (Pd-Ag, Pd-In) as well as monometallic Ag catalyst were reduced in $H_2$ flow at 550 °C for 1 h before the catalytic tests. Pd/$Al_2O_3$ catalyst was reduced in $H_2$ flow at 300 °C for 1 h. The catalyst was cooled down to the desired reaction temperature without $H_2$ flow termination and the propyne/p-$H_2$ mixture was introduced to the catalyst. The reactor was positioned outside an NMR magnet and the substrate gas mixture was supplied to the reactor and then the resulting mixture was supplied to the standard screw-cap 10-mm OD NMR tube (Merck, Z271969) placed inside NMR spectrometer for detection (ALTADENA experimental protocol, Pravica and Weitekamp, 1988) though a 1/16'' OD (1/32'' ID) PTFE capillary. In NMR tube the gas mixture was flowing from the bottom to the top and then to the vent through 1/4'' OD (5.8 mm ID) PTFE tubing connected with screw-ending of the NMR tube. All hydrogenation experiments were performed at ambient pressure (ca. 101 kPa). The reactor was heated with a tube furnace and the temperature was varied from 100 to 300 °C (in case of Pd-Ag catalyst) and to 500 °C (Pd-In catalyst) in 100 °C increments (heating rate was 10 °C/min). The temperature was controlled with a K-type thermocouple placed immediately adjacent to the catalyst bed on the external side of the reactor. The reaction was conducted in a continuous flow regime, with the reactor outflow continuously supplied to the probe of an NMR spectrometer and analyzed by [1]H NMR. The gas flow rate was controlled using an Aalborg rotameter and varied stepwise from 1.3 to 3.8 mL/s. The gas flow was periodically interrupted in order to acquire stopped-flow [1]H NMR spectra for evaluating conversion and selectivity. After the introduction of the substrate gas mixture to the catalyst and establishment of the steady-state regime, [1]H NMR spectra were recorded on a 300 MHz Bruker AV NMR spectrometer using a $\pi/2$ rf pulse. A 10-mm BBO 300 MHz Bruker probehead was used."

Questions on hydrogenation experiments:

3) In line 110 the authors mention that "propyne was premixed with p-$H_2$ in the molar ratio of 1:4". What was the overall pressure of the gas mixture before the reaction?

**Authors' response:**

In order to premix propyne with p-$H_2$, the gas tank was vacuumed (to ~ $10^{-4}$ atm) and propyne was introduced up to 1 atm pressure; then p-$H_2$ gas was added to the gauge pressure of 4 atm. However, the overall pressure in the gas tank is not relevant because, as stated in the manuscript, all hydrogenation experiments were performed at ambient pressure (ca. 101 kPa).

4) The authors performed the hydrogenation experiments at different temperatures, showing how temperature affects the conversion and selectivity of the reaction. It would be also interesting to know how different pressures would affect the reaction. Did the authors

ever tried to perform the hydrogenation with different pressures and can give a hint in which way the hydrogenation is affected by different pressures?

**Authors' response:**

This is a very interesting and important question, but also a very difficult one to answer at this time, at least in a simple and definite way.

There have been a few studies published in the past that address this very important issue of pressure effects in HET-PHIP. In particular, one study (Barskiy et al., 2017) addressed propene hydrogenation over $Rh/TiO_2$ catalysts under PASADENA conditions. It showed that the NMR signal enhancement observed for propane depends on both the overall pressure (1-7 bar) of reaction gas mixture (propene + p-$H_2$) and the propene fraction in the mixture (0-50%). The conversions and the intensity of polarized signals were higher at both higher overall pressures and higher propene fractions. At the same time, polarization levels (or signal enhancements) were decreasing with increasing propene fraction at constant overall pressure, but quickly passed through a pronounced maximum with the increase in the overall pressure at a constant propene fraction. Thus, signal enhancement was efficient only in a narrow range of relatively low propene fractions and total reaction pressures.

Another study (Salnikov et al., 2013) addressed the kinetics of propene hydrogenation over a $Pt/Al_2O_3$ catalyst. In this case, however, the experiments were performed at a constant overall pressure (1 bar) with He used as a balance gas. As a result, the reaction orders with respect to hydrogen were found to be different for the pairwise and the non-pairwise hydrogen addition and were equal to 0.7 and 0.1, respectively. It implies that while the overall conversion was essentially insensitive to p-$H_2$ pressure, the polarization level increased almost linearly with increasing p-$H_2$ pressure. It also implies that the contribution of pairwise addition depends on the fraction of molecular hydrogen in the mixture.

Therefore, the results of any variable pressure studies will largely depend on how exactly the experiment is performed, e.g., i) at constant overall pressure while varying the partial pressures of a substrate and p-$H_2$; ii) at constant overall pressure in the presence of a balance (inert) gas while changing partial pressures of one or both reactants; iii) with a variable overall pressure at a constant substrate:p-$H_2$ ratio, etc.

There are additional factors that would further complicate any such study and its interpretation. i) The reaction is highly exothermic, so an increased conversion (e.g., as a result of an increase in the overall pressure) will lead to the catalyst temperature rise, changing the rates of all kinetic stages of the reaction and possibly altering the catalyst structure. ii) If a catalyst is tested under the ALTADENA protocol, the decrease in the volume flow rate upon an increase in the gas pressure will increase the contact time between the reactants and the catalyst and can also increase the gas delivery time to the NMR, altering the reaction progress and boosting relaxation losses during gas transfer. iii) In both ALTADENA and PASADENA protocols, the situation is further exuberated by the $T_1$ relaxation time dependence on the overall gas pressure and gas mixture composition.

The effects of the reaction pressure on the catalyst activity in pairwise addition are the subject of our ongoing studies. Here in this work we studied catalytic behavior of different Pd-based

bimetallic catalysts in hydrogenation with parahydrogen under the identical reaction conditions, while the complex issue of pressure effects is definitely outside the scope of this work.

5) In table 1 the authors provide conversion rates and selectivity values for different flow rates of the gas mixture and state in line 124-125 that "Slightly higher selectivity values at high gas mixture flow rates is explained by the lower catalyst contact times". However, any information on the design of the tubular reactor for the hydrogenation reaction is missing (i.e. inner diameter and length of the reactor, length of fixed bed containing solid catalyst) that would allow the reader to estimate contact times of the gas stream with the catalyst bed from the provided flow rates.

**Authors' response:**

Regarding the design of the tubular reactor, a stainless steel tube with 1/4'' OD, 4.2 mm ID and 20 cm total length equipped with the stainless steel nuts and ferrules was used as the reactor. The catalyst was placed in the middle of the tube between two plugs of fiberglass tissue.
The catalyst loading ($m_{cat}$) used in the work is 30 mg, the apparent density of the granular catalyst bed ($\rho$) is 0.67 g·cm$^{-3}$, so the length of the catalyst bed can be evaluated as follows:

$$l = \frac{4 m_{cat}}{\rho \cdot \pi \cdot d^2} = 3.2 \ mm.$$

To a first approximation it can be assumed that the reaction proceeds in an ideal plug-flow reactor, so the contact time ($\tau$) can be crudely estimated as follows:

$$\tau = \frac{V_{cat}}{u_0} = \frac{m_{cat}}{\rho \cdot u_0},$$

where $u_0$ is the volume flow rate of the substrate/p-$H_2$ mixture. For 1.3 mL/s flow rate the contact time equals 34 ms; for 3.8 mL/s $\tau$ =12 ms.
However, it should be noted that the values presented above are, without doubt, underestimated since hydrocarbons are known to adsorb strongly on the surface of porous catalysts. For instance, direct observation of the adsorbed hyperpolarized products in HET-PHIP studies by MAS NMR have been reported previously (Henning et al., 2013). The pool of adsorbed reactants and products is constantly exchanging with the flowing gas, resulting in a significant increase in the effective contact time. The later will depend on the amount of adsorbed hydrocarbons, which in turn depends on pressure, gas mixture composition, catalyst temperature, etc.
It is thus essentially impossible to estimate a realistic value of the contact time.
However, there can be no doubt that an increase in the gas flow rate inevitably reduces the contact time between the gas and the catalyst bed. The direct consequence of this is the reduced reaction conversion at higher flow rates, the effect which is clearly observed experimentally. As a crude approximation, the contact time should change in the inverse proportion to the gas flow rate. However, strictly speaking this is valid only if the exchange between the gas and the adsorbed pool of molecules is rapid and complete, an assumption which in itself is difficult to ascertain.
Based on the arguments outlined above, we therefore believe that providing any estimates of the contact times in the manuscript would be misleading and inappropriate.

Questions on NMR experiments and relaxation issues

6) Line 111: Were the NMR experiments conducted as continuous flow experiments (i.e. on the flowing gas) (if so, please provide the mean velocity of the flowing sample) or in a stopped flow fashion? Was the used probehead a standard probehead for 5mm NMR tubes? How does the sample container in the NMR probehead looked like? Was it just a tube passed through the NMR coil (if so, please provide the inner diameter of the tube) or a flow cell with a more sophisticated geometry (if so, please provide a description of the geometry)? These points are important for a better understanding of the NMR experiments, as all points have an impact on the SNR of the spectra (i.e. filling factor, outflow effects and line broadening in continuous flow NMR).

**Authors' response:**

***The Materials and Methods section now contains the following information:***
The reaction was conducted in a continuous flow regime with the reactor outflow continuously supplied to the probe of an NMR spectrometer and analyzed by [1]H NMR. The PHIP spectra were acquired while the gas mixture was flowing. In order to evaluate the conversion and selectivity for each data point, the gas flow was periodically interrupted in order to acquire stopped-flow [1]H NMR spectra under thermal equilibrium conditions. After the introduction of the reactant gas mixture to the catalyst and after each change in the gas flow rate, we waited to ensure that the steady-state regime was established.
The reaction mixture from the reactor was supplied to the bottom of a standard screw-cap 10-mm NMR tube (Merck, Z271969) positioned in a 10-mm BBO 300 MHz Bruker probehead though a 1/32'' ID (1/16'' OD) PTFE capillary. The gas mixture was supplied via the capillary to the bottom of the NMR tube and was then flowing from the bottom to the top and on to the vent though 1/4'' OD (5.8 mm ID) PTFE tubing connected with screw-ending of the NMR tube.

***Further comments:***
Because the number of molecules in the gas phase decreases in the hydrogenation reaction (*propyne + H₂ → propene*), the gas volume flow rate after the reactor (the flow rate that determines the transport time of hydrogenation products and an unreacted substrate to the NMR tube) depends on the conversion:

$$u = u_0 \cdot \frac{5 - X}{5},$$

where $X$ is conversion value (between zero and one), $u_0$ is the inlet gas flow rate.
So, for example, for Pd catalyst tested at 200 °C and 3.8 mL/s flow rate, the conversion was 92 %, so the gas in the NMR tube is flowing with the following rate: $u = 3.8 \cdot \frac{5-0.92}{5} = 3.10$ mL/s. The volume of the sensitive region of rf coil is ~ 2 mL, so the residence time is ~0.65 s. This shortens an FID and leads to line broadening of NMR signals. As it was shown earlier (Barskiy et al., 2017), this situation is especially unfavorable in case of a PADADENA experiment, since antiphase multiplets will collapse significantly with line broadening – the flow rate of about 4 mL/s reduces PASADENA signals for hyperpolarized propane by a factor of ~2. Line broadening is much less critical for propene hyperpolarized in ALTADENA experiment since the chemical shifts for CH and CH₂-groups differ by ~ 0.8 ppm so that line broadening does not reduce the integrals of the polarized lines.

7) The distance between the tubular reactor and the RF coil of the NMR spectrometer should also be mentioned in the experimental section or even better the transport time (for the two different flow rates) of the hyperpolarized gas from the reactor to the detection site. These times together with the T1 relaxation time of propene (which should also be provided) would help the reader to estimate how severe hyperpolarization loss due to T1 relaxation was. This could also explain the very different SE values for the experiments conducted with 1.3 and 3.8 ml/s flow rates. The authors started a discussion on hyperpolarization losses due to T1 relaxation in chapter 3.3 (line 194-211), but do not provide enough information to understand if the hyperpolarization losses due to $T_1$ relaxation were dramatic (e.g. on the order of 60% of the initial signal enhancement) or minor (e.g. on the order of 10% of the initial signal enhancement).

**Authors' response:**
The difference in SE values observed at different flow rates is without doubt determined by the relaxation losses while the gas is flowing from the reactor to the NMR tube; this is an established fact.

1/32'' ID PTFE capillary was connected with the outflow end of the reactor; the other end of the capillary was inserted in the NMR tube all the way down to the bottom. The capillary length is ~ 240 cm. So, the capillary volume is $V_{capillary} = \frac{\pi \cdot d^2}{4} \cdot l = 1.19$ cm$^3$. The catalyst (3.2 mm length) was placed in the middle of a stainless steel tube (20 cm length, 4.2 mm ID), so the tube volume that the hyperpolarized products need to pass after contacting the catalyst is $V_{tube} = \frac{l \cdot \pi d^2}{4} = 1.38$ cm$^3$. The time-of-flight (ToF) of hydrogenation products depends on the conversion, $ToF = \frac{V_{capillary} + V_{tube}}{u} = \frac{V_{capillary} + V_{tube}}{u_0 \cdot \frac{5-X}{5}}$. For example, for Pd catalyst tested at 200 °C and 3.8 mL/s flow rate, ToF equals $\frac{1.19 + 1.38}{3.8 \cdot \frac{5 - 0.92}{5}} = 0.83$ s. This value is already comparable to the $T_1$ value of propene at the spectrometer field (ca. 0.6 s) (Skovpin et al., 2013). At the flow rate of 1.3 ml/s, ToF is ca. 2.47 s, and thus the relaxation losses should be dramatic.

These numbers provide the general picture of the overall situation, but a more or less accurate estimation of the relaxation losses from ToF values is not possible. The $T_1$ time of propene depends on pressure, gas mixture composition, temperature, etc. It also depends on the magnetic field in a non-trivial way (for instance, at low fields the long-lived spin states may be involved (Kovtunov et al., 2014)), and the gas experiences magnetic field with a varying amplitude while in transit.

Some useful information on polarization losses during the transfer of HP products can be found in some of our precious publications in which the "true" pairwise $H_2$ addition percentages were evaluated by taking into account relaxation losses and non-adiabaticity caused by fast gas inflow. Those studies demonstrated that the apparent signal enhancement factors can be significantly reduced due to the abovementioned causes – by one order of magnitude or possibly even more (Barskiy et al., 2015; Burueva et al., 2018).

We did not directly measure the polarization losses due to the transfer of HP products in this study because it is not essential for comparing different catalysts. The comparison is valid if the gas flow rate is the same in the experiments that are being compared, which ensures that relaxation losses are similar and thus the observed difference in polarizations is brought about solely by the use of different catalysts.

*We have added the discussion of hyperpolarization losses evaluation in section 3.3. This part now reads:*

"Hence, the apparent signal enhancements evaluated for Pd-based catalysts presented in Table 2 and Table 3 are lower estimates of the actual values of initial enhancements, underestimated (and possibly significantly) because hyperpolarization relaxation dramatically reduces the intensities of enhanced [1]H NMR signals of propene during the transfer. The accurate analysis of processes leading to polarization losses in the ALTADENA experiment (non-adiabaticity of magnetic field change during the transfer of hyperpolarized product from the Earth's magnetic field, the relaxation losses in both high and low magnetic fields, etc.) performed previously (Barskiy et al., 2015; Burueva et al., 2018) indicates that the apparent signal enhancement factors are significantly reduced due to the abovementioned causes – by one order of magnitude or possibly more. ….. At the same time, while minimization of relaxation losses is very important for applications of HET-PHIP, the primary objective of this work is the exploration of how the nature of a catalyst affects its selectivity toward pairwise $H_2$ addition to a substrate."

Technical comments:
Line 199: please correct the wrong spelling of "experimental"

**Authors' response:**
This typo has been corrected.

Line 265: add "in" before "order"

**Authors' response:**
The typo has been corrected. The document was proofread and the text was corrected where required.

**References**

Barskiy, D. A., Salnikov, O. G., Kovtunov, K. V. and Koptyug, I. V.: NMR signal enhancement for hyperpolarized fluids continuously generated in hydrogenation reactions with parahydrogen, J. Phys. Chem. A, 119, 996–1006, doi:10.1021/jp510572d, 2015.

Barskiy, D. A., Kovtunov, K. V., Gerasimov, E. Y., Phipps, M. A., Salnikov, O. G., Coffey, A. M., Kovtunova, L. M., Prosvirin, I. P., Bukhtiyarov, V. I., Koptyug, I. V. and Chekmenev, E. Y.: 2D Mapping of NMR Signal Enhancement and Relaxation for Heterogeneously Hyperpolarized Propane Gas, J. Phys. Chem. C, 121, 10038–10046, doi:10.1021/ACS.JPCC.7B02506, 2017.

Burueva, D. B., Kovtunov, K. V., Bukhtiyarov, A. V., Barskiy, D. A., Prosvirin, I. P., Mashkovsky, I. S., Baeva, G. N., Bukhtiyarov, V. I., Stakheev, A. Y. and Koptyug, I. V.: Selective single-site Pd-In hydrogenation catalyst for production of enhanced magnetic resonance signals using parahydrogen, Chem. - A Eur. J., 24, 2547–2553, doi:10.1002/chem.201705644, 2018.

Henning, H., Dyballa, M., Scheibe, M., Klemm, E. and Hunger, M.: In situ CF MAS NMR study of the pairwise incorporation of parahydrogen into olefins on rhodium-containing zeolites Y, Chem. Phys. Lett., 555, 258-262, doi:10.1016/j.cplett.2012.10.068, 2013.

Kovtunov, K. V., Truong, M. L., Barskiy, D. A., Koptyug, I. V., Coffey, A. M., Waddell, K. W. and Chekmenev, E. Y.: Long-lived spin states for low-field hyperpolarized gas MRI, Chem. - A Eur. J., 20, 14629–14632, doi:10.1002/chem.201405063, 2014.

Salnikov, O. G., Kovtunov, K. V., Barskiy, D. A., Bukhtiyarov, V. I., Kaptein, R. and Koptyug, I. V.: Kinetic Study of Propylene Hydrogenation over $Pt/Al_2O_3$ by Parahydrogen-Induced Polarization, Appl. Magn. Reson., 44(1–2), 279–288, doi:10.1007/s00723-012-0400-3, 2013.

Skovpin, I. V., Zhivonitko, V. V., Kaptein, R. and Koptyug, I. V.: Generating parahydrogen-induced polarization using imobilized iridium complexes in the gas-phase hydrogenation of carbon-carbon double and triple bonds, Appl. Magn. Reson., 44, 289–300, doi: 10.1007/s00723-012-0419-5, 2013.